# Evaluating *Beauveria bassiana* Strains for Insect Pest Control and Endophytic Colonization in Wheat

**DOI:** 10.3390/insects16030287

**Published:** 2025-03-10

**Authors:** Lulu Liu, Shiming Liu, Qingfan Meng, Bing Chen, Junjie Zhang, Xue Zhang, Zhe Lin, Zhen Zou

**Affiliations:** 1Institutes of Life Science and Green Development, School of Life Science, Hebei University, Baoding 071002, China; 2State Key Laboratory of Integrated Management of Pest Insects and Rodents, Institute of Zoology, Chinese Academy of Sciences, Beijing 100101, China; 3Jilin Provincial Key Laboratory of Insect Biodiversity and Ecosystem Function of Changbai Mountains, Forestry College, Beihua University, Jilin 132013, China; 4Institute of Forestry Engineering, Guangxi Eco-Engineering Vocational and Technical College, Liuzhou 545004, China; 5Engineering Research Center of Natural Enemies, Institute of Biological Control, Jilin Agricultural University, Changchun 130118, China

**Keywords:** *Beauveria bassiana*, insecticidal activity, endophytic colonization, lepidopteran pests, biological control

## Abstract

Entomopathogenic fungi have long been regarded as a crucial biological control method. We studied the use of *Beauveria bassiana* to control insect pests that damage wheat crops. Three *B. bassiana* strains were tested for their ability to kill lepidopteran insects. Our results showed that all the strains were effective at causing a high mortality rate of pests in a short period of time, with one strain (CBM1) proving the most potent. Then, the ability of CBM1 to colonize wheat plants was examined. When applied to wheat seeds, *B. bassiana* helped the plants grow taller and develop longer roots. This research suggests that *B. bassiana* could be a valuable tool for protecting wheat crops from pests, potentially reducing the need for chemical pesticides and supporting sustainable farming practices.

## 1. Introduction

Wheat serves as the primary food source for approximately 35% of the global population, providing an amount of protein equivalent to that found in meat, eggs, and dairy products. Therefore, the health of the wheat industry is critically linked to food security and global stability. Challenges such as increasing yield, improving quality, enhancing disease resilience, and reducing production costs are major concerns to the agricultural sector. In recent years, global climate change, the invasion of pests like the fall armyworm (*Spodoptera frugiperda*), and declining agricultural biodiversity have led to more frequent and severe pest outbreaks [1]. Additionally, reports have indicated that species such as *Ostrinia nubilalis* and *Helicoverpa armigera* are significant worldwide threats to wheat, corn, and rice production due to their wide host range, high reproductive capacity, and adaptability [2,3]. Insect outbreaks can lead to substantial reductions in yield or even total crop loss within a short period, causing severe economic injury to agricultural production.

Lepidoptera is the largest group of plant-feeding insects, and its immune function has been extensively studied [4,5,6,7]. Although physical and chemical control methods are commonly used in agriculture, biological controls are increasingly recognized for their environmentally friendly characteristics and importance in integrated pest management [8]. The current widely employed biological insecticides include natural enemies and insect pathogenic microorganisms. These include predatory natural enemies such as the hoverfly *Episyrphus balteatus* [9], parasitic wasps [10,11,12], entomopathogenic fungi [13,14,15], entomopathogenic bacteria and viruses [16,17,18], and entomopathogenic nematodes [19,20]. Entomopathogenic fungi have long been regarded as a crucial biological control method, accounting for approximately 60% of the total mortality caused by all pathogenic microorganisms. Most research on fungal pathogens has focused on *B. bassiana* of the family Cordycipitaceae and *Metarhizium brunneum* of the family Clavicipitaceae. Based on these fungal strains, many commercially registered fungal insecticide products have been developed. Studies have demonstrated that *B. bassiana*, *M. anisopliae*, *Isaria fumosorosea*, and *B. brongniartii* are highly effective against a range of pests, including *Lepidoptera*, *Coleoptera*, *Hemiptera*, and *Diptera* [21]. However, under natural conditions, these entomopathogenic fungi exhibit a high level of host adaptability, leading to a prolonged infection period as they proliferate within the host before causing host death. While this characteristic helps increase fungal reproduction, it also limits the efficiency of field control. To overcome this limitation, genetic engineering techniques have been applied to the genomes of entomopathogenic fungi. Researchers aim to enhance fungal virulence by inserting exogenous genes or modifying the expression patterns of endogenous genes [22,23,24]. However, genetically engineered products still face challenges and strict regulations in practical applications due to their instability and concerns regarding ecological security.

Endophytes are microorganisms that live within plant tissues and significantly influence plant growth, stress resistance, and pest resistance. Entomopathogenic fungi, as a type of endophyte, can directly and indirectly affect plant physiological and biochemical processes. Such fungi not only play a role in regulating insect populations but also enhance the plant’s intrinsic defense mechanisms, providing new strategies for the long-term management of plant pests and diseases. *M. anisopliae* and *B. bassiana* have been the most extensively studied species among the entomopathogenic fungi. Various inoculation methods, such as spraying, injection, seed immersion, and soil drenching, have successfully established fungal colonization in numerous plants, including *Triticum aestivum* [25,26], *Gossypium hirsutum* [27], *Solanum lycopersicum* [28], *Zea mays* [29,30], *Solanum tuberosum* [31], *Nicotiana benthamiana* [32], *Phaseolus vulgaris* [33], and *Vitis vinifera* L. [34]. Compared to traditional entomopathogenic fungal agents, endophytic fungi offer the advantage of establishing close relationships with host plants either naturally or through artificial inoculation, thus being less susceptible to environmental stresses.

To address potential pest issues before they arise, a pre-emptive biological control strategy can be employed, including releasing natural predators in the field, applying fungal insecticides ahead of time, and developing and planting crops with built-in pest resistance [35,36,37]. By anticipating potential challenges and addressing these in advance, farmers can reduce risk and improve the overall resilience and productivity of their crops. To this end, our study identified three novel strains of *B. bassiana* that had not been previously reported. Insecticidal activity assays on lepidopteran insects led to the selection of *B. bassiana* CBM1 as the most potent strain. Successful fungal colonization in wheat was achieved using different inoculating methods, resulting in high plant colonization rates. The wheat produced not only grew better but also exhibited stronger resistance to insect pest damage. This study demonstrates the potential of *B. bassiana* strains as a pre-emptive tool in strategies for biocontrol in wheat fields.

## 2. Materials and Methods

### 2.1. Insects and Fungi

The tested insects included *H. armigera*, *S. frugiperda*, *M. separate*, and *P. xylostella.* All insect larvae were obtained from laboratory populations. The larvae of *H. armigera*, *S. frugiperda*, and *M. separata* were fed either an artificial diet free of preservatives or fresh wheat (*T. aestivum* L.) seedlings that were grown hydroponically, while *P. xylostella* was provided with fresh cabbage. All insects were reared in an artificial incubator set at 26 ± 1 °C with a 14 h light and 10 h dark cycle (14 L:10 D) with a relative humidity (RH) of 60%.

The three fungal strains used in the experiments were isolated and purified from larvae of *Curculio dentipes* and adults of *Massicus raddei* and *Mohechotypa diphysis* collected from the Changbai Mountains. The fungal strains were named CBM1, CBM2, and CBM3 based on their collection location. Strains of CBM1 and CBM3 have been deposited in China General Microbiological Culture Collection Center (CGMCC), and the access numbers are CGMCC No.41811 and CGMCC No.41812, respectively. The strains were inoculated onto potato dextrose agar (PDA) and cultured at 26 °C. After the conidial suspension was applied to the plates or mycelial inoculum was introduced, the next generation of conidia could be harvested after approximately 20 days. The conidia were used to prepare spore suspensions or were stored in 30% glycerol at −80 °C for future use.

### 2.2. DNA Extraction, PCR Amplification, and Sequencing

The three fungal strains were cultured on PDA plates at 26 °C in the dark for seven days. After the incubation period, the mycelium was scraped off the plates using a sterile spatula. Genomic DNA was extracted from the fungi using an Ezup Column Fungi Genomic DNA Purification Kit (Sangon, Shanghai, China).

The ITS regions of each strain were amplified by PCR and sequenced using the extracted genomic DNA as a template. The universal primers ITS1 (5′-TCCGTAGGTGAACCTGCGG-3′) and ITS4 (5′-TCCTCCGCTTATTGATATGC-3′) were used [38,39]. The PCR amplification procedure was as follows: 30 s at 98 °C, followed by 34 cycles of 10 s at 98 °C, 5 s at 54 °C, 5 s at 72 °C, and a final extension of 1 min at 72 °C. The PCR products were sent for sequencing after confirming the correct size through 1% agarose gel electrophoresis.

The sequencing data have been uploaded to Science Data Bank (DOI: 10.57760/sciencedb.13591). The sequencing results were compared with homologous gene fragments of related species downloaded from the NCBI database using BLAST (https://blast.ncbi.nlm.nih.gov/Blast.cgi, last accessed on 13 October 2024), and a homology analysis was conducted. Multiple sequence alignments were performed using MEGA 11, and a phylogenetic tree was constructed using the neighbor-joining (NJ) method with bootstrapping conducted to calculate the credibility of the branches using 1000 replicates [40].

### 2.3. Insecticidal Activity of Entomopathogenic Fungi

#### 2.3.1. Preparation of Conidial Suspensions

The total mycelium from fungal plates incubated for approximately three weeks was scraped into a 50 mL centrifuge tube using a sterile spatula in a laminar flow cabinet. The material was rinsed with sterile water and vortexed for five minutes to thoroughly dislodge the conidia from the mycelium. The resulting suspension was filtered through a sterile cotton-plugged syringe to remove any remaining mycelial debris, yielding a conidial suspension. Spore morphology and concentration were assessed using a 25 × 16 mm hemocytometer.

Two concentrations of spore suspensions (1 × 10^8^ and 1 × 10^7^ conidia/mL) were prepared for each of the three strains CBM1, CBM2, and CBM3. Then, 0.05% Tween-80 was added to the spore suspensions (treated) or sterile water (control) before use.

#### 2.3.2. Insecticidal Activity Assay

The insecticidal activities of CBM1, CBM2, and CBM3 were assessed by soaking larvae in spore suspensions. For each isolate, second-instar larvae of *H. armigera*, *S. frugiperda*, and *M. separata* and fourth-instar *P. xylostella* larvae were individually placed in a 12-well cell culture plate containing freshly cut wheat segments to prevent cannibalism. Each larva was immersed in the corresponding spore suspension or without conidia (control) for 15 s, and excess droplets were wiped off. Spore concentrations of 1 × 10^8^ and 1 × 10^7^ conidia/mL were used for initial insecticidal assessment. Each treatment had three replicates with 12 larvae, and the experiments were repeated three times. After infection, the larvae were maintained in the same rearing incubators. Food was replenished daily, and dead larvae were removed and recorded. The dead larvae were disinfected with 1% sodium hypochlorite (NaOCl) for one minute followed by two rinses with sterile water and then transferred to plates to observe fungal colony growth. The experiments were repeated three times.

#### 2.3.3. Calculation of LC_50_ and LT_50_

The LC_50_ (median lethal concentration) and LT_50_ (median lethal time) values were determined using the soaking method. Based on the preliminary tests, conidial suspensions of varying concentrations were prepared for each insect against each strain. For the test of *P. xylostella*, *H. armigera*, and *S. frugiperda*, 1 × 10^8^, 5 × 10^7^, 1 × 10^7^, 5 × 10^6^, and 1 × 10^6^ conidia/mL conidial suspensions of *B. bassiana* CBM1 and CBM3, and 1 × 10^9^, 5 × 10^8^, 1 × 10^8^, 5 × 10^7^, 1 × 10^7^, 5 × 10^6^, and 1 × 10^6^ conidia/mL conidial suspensions of CBM2 were used. For the test of *M. separata*, 1 × 10^9^, 1 × 10^8^, 1 × 10^7^, and 1 × 10^6^ conidia/mL conidial suspensions of CBM1 and CBM3 and 1 × 10^9^, 1 × 10^8^, and 1 × 10^7^ conidia/mL conidial suspensions of CBM2 were used. Each treatment had three replicates with 12 larvae, and the experiments were repeated three times.

### 2.4. Entomopathogenic Fungal Colonization of Wheat

#### 2.4.1. Plant Resources and Cultivation Conditions

The wheat used in this study was Jimai 22 (National Approval Original Wheat Seed 2006018). This variety is known for its large grains, strong tillering ability, and good cold resistance. All wheat plants were grown in sterilized potting soil (a 1:1 mixture of floral soil and vermiculite). Two germinated seeds were planted per 8 × 8 × 8 cm (length × width × height) pot. The plants were maintained in a greenhouse under the following environmental conditions: a temperature of 28 ± 1 °C, RH of 60% ± 10%, and a 14 h light and 10 h dark cycle.

#### 2.4.2. Seed Immersion Treatment

The isolate *B. bassiana* CBM1 was used to inoculate wheat seeds using suspensions at a concentration of 1 × 10^8^ conidia/mL containing 0.05% Tween-80. Seeds were sterilized by soaking in 3% NaClO and 70% ethanol for 3 min each, followed by three 45 s washes in sterile water. The seeds were then immersed overnight (10 h) in 30 mL conidial suspensions with continuous shaking at 180 rpm to ensure thorough contact between the spores and the seeds. After inoculation, the seeds were dried for 20 min in sterile Petri dishes. The seeds were then planted in pots 3 cm below the soil surface, and the time was referred to as 0 days after sowing (0 DAS). Each pot contained two seeds. Control seeds were treated in the same manner but with only sterile 0.05% Tween-80 solution [25,41]. After planting, all pots were placed in a 120-mesh insect-proof cage (63 cm × 50 cm × 45 cm) to prevent pest infestation and cross-contamination.

#### 2.4.3. Soil Drench Treatment

Wheat seeds were soaked in sterile water for one day and then drained and covered with sterilized damp towels to facilitate germination. Well-germinated seeds were then planted about 3 cm deep in the soil of pots (0 DAS) with two seeds per pot. After three days (3 DAS) when the seedlings had emerged, they were subjected to inoculation by a soil drench treatment. Thirty milliliters of *B. bassiana* CBM1 spore suspension (1 × 10^8^ conidia/mL) containing 0.05% Tween-80 was used to irrigate the soil for five consecutive days. The control group received irrigation with 0.05% Tween-80 sterile water only. After five days, the pots were watered as needed, and no fertilizer was applied.

#### 2.4.4. Assessment of Fungal Colonization

Samples from the seed immersion treatment group were collected at 6 DAS, while samples from the soil drench treatment were collected at 12 DAS, 5 days after the completion of soil drenching. The method for surface sterilization of the samples followed Greenfield et al. [42]. The wheat leaves were rinsed with sterile distilled water to remove surface soil and debris and then disinfected with 0.5% NaClO for 1 min, followed by 70% ethanol for 30 s. The leaves were subsequently rinsed three times with sterile water and then dried with sterile paper. Leaf tissues were cut into 1 cm segments and placed on PDA plates using sterile tweezers. Nine segments were randomly arranged in a 3 × 3 grid on each plate, with four replicates per plant. Twenty plants were set to calculate the fungal colonization rate in each experiment, and the experiments were repeated three times. All plates were then sealed with parafilm and incubated in the dark at 26 °C.

The growth of fungal colonies on the plant tissues was assessed after seven days of incubation. The number of plates displaying fungal growth was recorded, and the fungal colonization rate was calculated [28,43]. Colonization rate = (number of plates with colonies/total number of sampled plants) × 100%.

### 2.5. Effect of Fungal Colonization on Plant Growth

This experiment was performed using the soil drench method. Plants in the treated group were inoculated by *B. bassiana* CBM1 at a concentration of 1 × 10^8^ conidia/mL (containing 0.05% Tween-80), as per the method in Section 2.4.4 described above. The control group was treated in the same manner but with only sterile 0.05% Tween-80 solution. Plant height was measured with a ruler 10 times within thirty days, and root length was measured 7 times within fifteen days, until the data exhibited an asymptotic curve [26]. For assessment of plant growth, a randomized block design was used with six plants for each experiment. The experiment was repeated three times.

### 2.6. S. frugiperda Feeding with CBM1-Colonized Wheat

The CBM1 strain that yielded the highest mortality rate was used to inoculate the plants via soil drench to achieve extensively colonization of wheat leaves. The leaves were chopped and placed in 12-well cell culture plates to serve as food for the insects. The second-instar *S. frugiperda* larvae that had been starved for 24 h were then introduced into the 12-well plates with one larva per well and were fed with colonized leaves. For the control group, fresh fungi-free wheat leaves were used. Each treatment had three replicates with 12 larvae, and the experiments were repeated three times. Insect growth, development, and mortality were recorded daily, and any dead larvae were promptly removed from the plates.

### 2.7. Statistical Analysis

Before analysis, data were subjected to normality and homogeneity tests of variance. Survival curves were plotted using Kaplan–Meier analysis [44] in GraphPad Prism 9.0 or by https://www.bioinformatics.com.cn (last accessed on 21 November 2024), an online platform for data analysis and visualization. Differences in insect survival among the three strains of different concentrations were estimated by a Log-rank (Mantel–Cox) test. One-way analysis of variance (ANOVA) was used to analyze the mortality data for insects and growth parameters for plants, and the *t*-test was used for colonization rate comparison. Post hoc comparison was performed using LSD and Duncan’s test when significant main effects were detected (*p* < 0.05). The LT_50_ values and LC_50_ values were calculated using PROBIT analysis provided by SPSS Statistics 26.0.

## 3. Results

### 3.1. Identification of Three B. bassiana Isolates

The colonies of all strains exhibited typical characteristics of *B. bassiana*. The colonies were creamy white or pale yellow, with central conidiation areas ranging from fluffy to powdery. Colonies of CBM1 and CBM2 appeared round and flat, while CBM3 displayed a visible fluffy mycelium. The colony margins extended radially with a fluffy mycelial appearance. After two weeks of incubation, round or oval, transparent, smooth conidia were visible under an optical microscope. Sequence analysis revealed that the ITS regions of CBM1, CBM2, and CBM3 were 530 bp in length, excluding the primers (Figure 1A). Blasting of the ITS sequence of the three novel strains and other fungi followed by sequence alignment and phylogenetic tree construction showed that CBM1, CBM2, and CBM3 had a very close relationship to the *Beauveria* genus (Figure 1B). Combining the molecular analysis with morphological observation confirmed that the three isolates were *B. bassiana*.

### 3.2. Bioassays of B. bassiana Strains CBM1, CBM2, and CBM3 Against Lepidoptera

The treatments with 1 × 10^8^ and 1 × 10^7^ conidia/mL conidial suspensions of *B. bassiana* isolates CBM1, CBM2, and CBM3 significantly reduced the survival rates of *H. armigera* (Log-rank test: χ^2^ = 335.3, *df* = 6, *p* < 0.0001, Figure 2A), *M. separata* (Log-rank test: χ^2^ = 236.3, *df* = 6, *p* < 0.0001, Figure 2B), *S. frugiperda* (Log-rank test: χ^2^ = 141.8, *df* = 6, *p* < 0.0001, Figure 2C), and *P. xylostella* (Log-rank test: χ^2^ = 229.7, *df* = 6, *p* < 0.0001, Figure 2D) compared to those in the control. In the virulence bioassays with *H. armigera*, there were significant differences in cumulative mortality rates at the fourth day post infection at the spore concentration of 1 × 10^7^ conidia/mL (F = 103.453, *p* < 0.05). Dead larvae were disinfected and transferred to PDA plates for incubation; white colonies grew from the larvae after seven days (Appendix A). The morphology of the colonies resembled the characteristics of *B. bassiana* CBM1 used for infection, indicating that the fungal spores had penetrated the insect’s cuticle and were the cause of larval mortality.

To further determine the pathogenicity of *B. bassiana* strains, more comprehensive experiments were conducted with various spore concentrations against *P. xylostella* fourth-instar larvae and *S. frugiperda*, *H. armigera*, and *M. separata* second-instar larvae to calculate the LT_50_ (Table 1) and LC_50_ values (Table 2). The results showed that at a spore concentration of 1 × 10^8^ conidia/mL, *B. bassiana* CBM1 had the lowest LT_50_ value, with CBM2 having a higher LT_50_ than CBM3. For the LC_50_ comparisons, at seven days post infection, the LC_50_ for *B. bassiana* CBM1 was only 0.54-fold that of CBM2, with CBM3 showing a value that was 0.61-fold that of CBM2. As a consequence, *B. bassiana* CBM1 was the most lethal strain, followed by CBM3 with intermediate virulence and CBM2 the least.

### 3.3. Comparison of the Colonization Rate of B. bassiana CBM1 in Wheat by Seed Immersion and Soil Drenching

We identified *B. bassiana* CBM1 as having strong pathogenicity against lepidopteran pests through the above experiments. This strain was then used to study fungal colonization in wheat plants.

For the seed immersion treatment, wheat seeds were soaked for 10 h in a conidial suspension and then germinated and planted in a greenhouse. Leaf samples were collected at 6 DAS. For the soil drench treatment, the fungi were inoculated continuously from 3 to 7 DAS, and leaf samples were collected on the 5th day after the final inoculation (12 DAS). After seven days of incubation, no colonies were observed on the PDA inoculated with sterile water from the final leaf wash (Figure 3A), indicating that surface sterilization of the wheat leaves was effective and that no contaminants were present. Compared to the leaves from the control group (Figure 3B), white colonies consistent with *B. bassiana* CBM1 morphology were observed on the PDA plates for plant tissues from both the seed immersion and soil drench treatment experiments (Figure 3C). The hyphae and conidia of *B. bassiana* could be observed under a microscope after diluting the colony grown from the wheat leaves (Figure 3D).

To compare the colonization rates between the two fungal inoculation methods, it was considered colonized if at least one piece showed fungal growth among nine tissue cuts on a plate. The resulting colonization rates are shown in Figure 3E. The colonization rate for the wheat plants treated by seed immersion with *B. bassiana* CBM1 was 42.92%, while the rate for the soil drench treatment was 81.25%, with a highly significant difference (T = 18.40, *df* = 4, *p* < 0.0001). This difference was speculated to be due to the continuous five-day inoculation that allowed the wheat seeds to root, germinate, and develop leaves during the treatment period, facilitating better interaction between the fungi conidia and the root or stem surface and promoting fungal endophytic growth within the wheat.

### 3.4. Effect of B. bassiana CBM1 Colonization on Plant Height and Root Length by Soil Drenching

Given that the soil drench method significantly enhanced the colonization of *B. bassiana* CBM1 in wheat plants with a colonization rate reaching 81.25%, we compared the height and root length of the plants between the treated and control groups. The results are shown in Figure 4. The height of the wheat plants treated with fungi was consistently higher than that of the untreated plants over 30 days (Figure 4B). Height differences between the two groups were compared on the same day. Significant increases in the treated group were observed at 8 (F = 25.142, *p* = 0.007), 12 (F = 26.988, *p* = 0.007), 16 (F = 46.276, *p* = 0.002), 20 (F = 54.831, *p* = 0.002), and 30 (F = 36.127, *p* = 0.004) DAS. The changes in root length of the fungal colonization and control groups were compared over 15 days (Figure 4D). Although there was a negative effect on root length in the colonization group at 4 and 5 DAS, starting from 7 DAS, the root lengths in the treated group were consistently greater than those in the control group, indicating a promoting effect. No significant differences were observed from 4 to 10 DAS, but the root length of colonization plants had a significant increase at 13 (F = 23.523, *p* = 0.008) and 15 (F = 12.877, *p* = 0.023) DAS. These results suggest that *B. bassiana* CBM1 not only enhances plant height but also promotes root growth in wheat.

### 3.5. Effect of Feeding S. frugiperda with B. bassiana CBM1-Colonized Wheat

Wheat leaves colonized by *B. bassiana* CBM1 were used to feed second-instar fall armyworm larvae. After 14 days, the cumulative mortality was compared between the treated and control groups (Figure 5A). The mortality of the larvae fed with CBM1-colonized wheat reached 52.78%, significantly higher than the 2.78% observed in the control group (F = 162.13, *p* < 0.0001), indicating that the *B. bassiana* CBM1-colonized wheat was effective against the larvae.

## 4. Discussion

Substantial research has demonstrated that *B. bassiana* is a highly effective biological control agent against agricultural pests [45,46,47]. However, its application in the field is often constrained by environmental factors. A connection between the strain and the host can be established by artificially inoculating *B. bassiana* onto host plants. This approach can not only mitigate environmental stress on *B. bassiana* but also offers new strategies for pest management in agricultural fields. Prior to this study, Behie et al. [48] observed that *B. bassiana* could colonize all plant tissues, including roots, hypocotyls, stems, and leaves.

Developing novel entomopathogenic fungal strains for use as biological insecticides and determining their ability for colonization in plant tissues are important goals in insect pest management. In this study, *B. bassiana* CBM1, CBM2, and CBM3 were isolated from *C. dentipes*, *M. raddei*, and *M. diphysis* living in cold environments, respectively. The insecticidal activities of these strains were evaluated against lepidopteran larvae of *H. armigera*, *M. separata*, *S. frugiperda*, and *P. xylostella* at various concentrations. *B. bassiana* F-HY006 was isolated from the cadavers of *Diaphorina citri*, and a concentration of 1 × 10^7^ conidia/mL achieved a corrected mortality rate of 76% and an LT_50_ of 4.09 days [49]. The pathogenicity of the two fungal strains *B. bassiana* 88 and *M. anisopliae* 129 was evaluated by spraying their conidial suspensions onto leaves. The results showed that both strains caused up to 93% mortality in adult *Brevipalpus yothersi*, significantly higher than the 12% mortality in the control group [50]. A study comparing the virulence of seven *B. bassiana* isolates against *S. frugiperda* larvae indicated that isolate Bb9 achieved 100% mortality at 1 × 10^8^ conidia/mL toward 24 h neonate larvae, whereas isolate Bb40 only achieved 19% mortality [51]. In our study, *B. bassiana* strains CBM1, CBM2, and CBM3 also achieved 100% mortality against second-instar larvae at 1 × 10^8^ conidia/mL. This high mortality rate, even in older larvae, demonstrates the virulence of the three isolated strains. As insects grow, their increased size and stronger immune responses increase the resistance to pathogens. Additionally, the microorganisms present on the insect cuticle can inhibit the germination and growth of fungal spores, thus reducing pathogenicity [52]. Therefore, using older larvae in screening entomopathogenic fungi as insecticides may be more effective, and applying these insecticides as early as possible in practical application is advisable.

The present study identified *B. bassiana* isolate CBM1 as having the strongest insecticidal activity and demonstrated its ability to colonize wheat plants through artificial inoculation. A comparison of colonization rates between seed immersion and soil drench methods revealed that under soil drenching, the colonization rate of *B. bassiana* in wheat leaves reached 88.75% five days after treatment, significantly higher than the rate achieved by seed immersion. The study further examined the impact of *B. bassiana* endophytic colonization on wheat growth, revealing that *B. bassiana* CBM1 significantly improved plant height and root length, with the treated plants showing increased height at 16 and 30 DAS and longer roots at 15 DAS compared to the control. The growth-promoting effect initiated by root inoculation of conidia influenced the overall growth of the wheat over time and also significantly increased the mortality of fall armyworm larvae. Related research indicates that fungal colonization preferences can vary among different plant tissues. For example, when nine *B. bassiana* strains were introduced into *Solanaceae* crops via root drenching, the highest colonization rate was observed in the stems [53]. *B. bassiana* GHA exhibited epiphytic growth on intact tomato surfaces under high humidity, and it established endophytic colonization near the inoculation sites, with increased frequency following surface scarification of plant surfaces [54]. Additionally, *B. bassiana* H2S32 colonizing the grapevine *Vitis vinifera* showed higher fungal colonization rates in the stems compared to the leaves, stems, and buds 53 days post inoculation [55].

Fungal colonization in plants can enhance plant growth. For example, introducing *B. bassiana* into *Vicia faba* and *T. aestivum* L. through seed inoculation has been shown to improve various growth parameters, including shoot height, root length, and fresh root and shoot weights [56,57]. Inoculating *B. bassiana* G41 into bananas using root and rhizome dipping significantly reduced plant damage and decreased the survival rate of *Cosmopolites sordidus* after feeding on the plants [58]. Moreover, colonization of plants with *B. bassiana* negatively affected pest species such as *H. armigera*, *Ophiomyia phaseoli*, *S. littoralis*, *Cameraria ohridella*, and *Myzus persicae*, leading to mortality or adverse effects on growth, development, reproduction, and behavior [33,59,60,61].

A previous study initiated *B. bassiana* endophytic colonization by spraying method. The fungus was re-isolated from the plant and inoculated into other plants again, and this was repeated on three plant species. Each re-isolated fungus was tested for insecticidal activity against *Galleria mellonella* L. fourth-instar larvae using a soaking method. All treatments resulted in mortality ranging from 73.3% to 100%, indicating that the virulence of *B. bassiana* remained stable after colonization and re-isolation [62]. This suggests that the insecticidal potential of entomopathogenic fungi can be stabilized through endophytic colonization, as opposed to colonization in artificial media, where potency often diminishes with successive passages.

Researchers have also explored the combined colonization of plants by multiple entomopathogenic fungi or the co-application of entomopathogenic fungi with various compounds. For example, *B. bassiana* ESALQ 3375 and *M. robertsii* ESALQ 1622 were tested individually and in combination on *Phaseolus vulgaris* after seed treatment, with the result being that all treatments suppressed the spider mite *Tetranychus urticae* and promoted the growth of bean plants; under the given test conditions, co-inoculation of both fungal isolates provided no additional benefit compared to applying each isolate individually [63]. Additionally, the thermoregulatory abilities of insects can affect the pathogenicity of entomopathogens. For example, the susceptibility of *Locusta migratoria* to *B. bassiana* increased following phenidone treatment, indicating that combining *B. bassiana* with phenidone can enhance fungal virulence [64]. The combined use of plant extracts and fungi has at times yielded better pest management results than chemical insecticides. A study applying orange oil extract with *B. bassiana* to rice *Oryza sativa* L. fields demonstrated that this combined treatment resulted in the smallest insect pest populations, and the *B. bassiana* and orange oil extract treatment achieved a favorable equilibrium between insect pest populations and their natural enemies [65]. Researchers have also investigated the effects of combining entomopathogenic fungi with natural enemies. When two predatory mites were each inoculated along with *B. bassiana* or *M. anisopliae* and then used against the mite *Brevipalpus yothersi*, the results showed that the combination of *Amblyseius swirskii* with the fungus *M. anisopliae* was the most promising biocontrol strategy [50].

In addition to the aforementioned benefits, endophytic colonization with fungi can suppress plant diseases by enhancing plant resistance to pathogens such as Verticillium wilt, gray mold, *Pythium myriotylum*, and *Rhizoctonia* diseases [53,66,67,68,69] While extensive research has been conducted on the endophytes of *B. bassiana*, including its growth-promoting, insecticidal, and disease-resistant properties, the effectiveness of these entomopathogenic-fungi-based biopesticides can be limited by factors such as temperature and humidity. Future strategies could involve combining biopesticide formulations with endophytic fungi and other agricultural control methods to enhance their efficacy. Pre-emptive application of fungi-colonized plants for crop production before pest and disease outbreaks has significant potential in the field of biological control.

## Figures and Tables

**Figure 1 insects-16-00287-f001:**
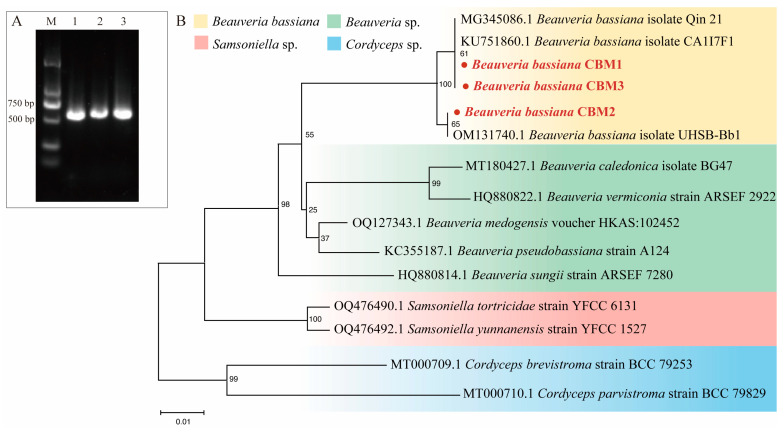
Molecular identification of three isolates of *Beauveria bassiana*. PCR amplification products of ITS region of three isolates (**A**). M: DNA marker 2000; 1: CBM1; 2: CBM2; 3: CBM3. Specific bands were observed between 500 bp and 750 bp. Phylogenetic tree based on the ITS sequence of novel isolates CBM1, CBM2, and CBM3, along with other homologous strains (**B**). The tree was constructed using the NJ method by MEGA 11, with identified strains highlighted in red. The scale bar represents 1% sequence divergence.

**Figure 2 insects-16-00287-f002:**
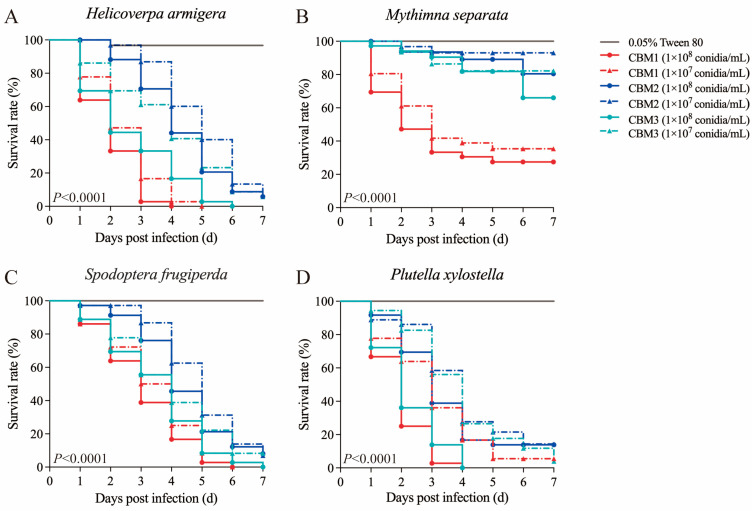
Kaplan–Meier survival curves of Lepidoptera larvae immersed in spore suspensions of *B. bassiana* isolates CBM1, CBM2, and CBM3 at different concentrations (1 × 10^8^ and 1 × 10^7^ conidia/mL). Tested insects: (**A**) *H. armigera*, (**B**) *M. separata*, (**C**) *S. frugiperda*, and (**D**) *P. xylostella*. Each group contained 12 larvae in triplicate. Significant differences were observed among the different treatments using the Log-rank test (*p* < 0.0001).

**Figure 3 insects-16-00287-f003:**
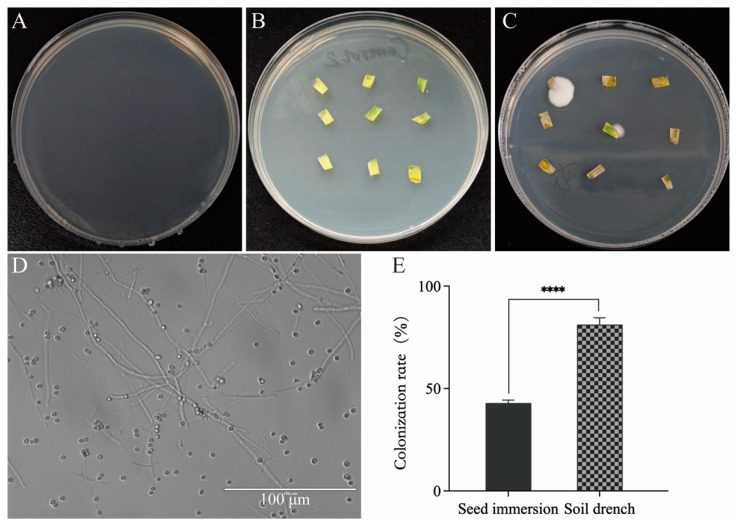
Assessment of colonization rate of *B. bassiana* CBM1 in wheat leaves. (**A**) No fungal colonies were observed on PDA medium inoculated with water from the final wash of leaf samples, confirming effective surface disinfection. (**B**) No fungal colonies were observed from the control group when leaf samples were inoculated onto PDA medium. (**C**) White colonies with morphology consistent with *B. bassiana* CBM1 were observed from the treated group. (**D**) Hyphae and conidia of *B. bassiana* CBM1 were observed under a microscope after diluting the colonies from the treated group (**D**). (**E**) For the seed immersion and soil drench treatments, leaf samples were plated on PDA and incubated for seven days to assess colonization rates. Twenty plants were set to calculate the fungal colonization rate in each group, and the experiments were repeated three times. Values are mean ± SD. **** indicates significant differences between groups as determined by the *t*-test (*p* < 0.0001).

**Figure 4 insects-16-00287-f004:**
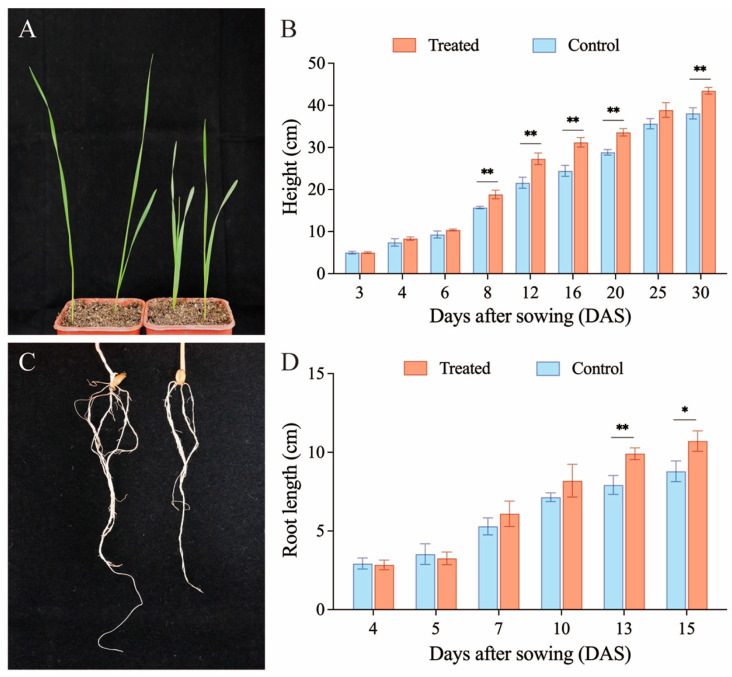
The effect of *B. bassiana* CBM1 colonization on plant height and root length by soil drenching. The left plants are from the soil drench treatment group, and those on the right are from the control group (**A**,**C**). Treated group refers to the soil drench colonization group, and the control group refers to the group without fungal colonization. Six plants were used in each group, and the experiments were repeated three times. Values are mean ± SD. Significant differences in plant height (**B**) and root length (**D**) were observed at specific days between groups based on one-way ANOVA (*p* < 0.05). * represents *p* < 0.05, ** represents *p* < 0.01.

**Figure 5 insects-16-00287-f005:**
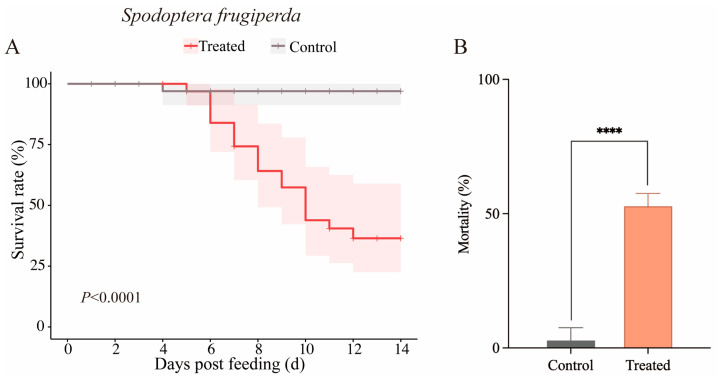
Kaplan–Meier survival analysis of *S. frugiperda* larvae fed with *B. bassiana* CBM1-colonized wheat leaves via the soil drench method and comparison of cumulative mortality between the treated and control groups. Each group contained 12 larvae in triplicate. The Log-rank test results were as follows: χ^2^ = 14.58, *df* = 1, *p* < 0.0001 (**A**). The cumulative mortality of *S. frugiperda* larvae fed with *B. bassiana* CBM1-colonized wheat after 14 days also showed a highly significant difference, as determined by one-way ANOVA (*p* < 0.05) (**B**). **** indicates significant differences between the treated and control groups.

**Table 1 insects-16-00287-t001:** LT_50_ of three *B. bassiana* isolates to four Lepidoptera larvae. LT_50_ refers to the lethal time to 50% mortality, calculated for soaking treatment with concentrations of 1 × 10^8^ and 1 × 10^7^ conidia/mL. Each group contained 12 larvae and was performed in triplicate.

Concentration	Strains	LT_50_ (d) (95% Confidence Interval)			
		*P. xylostella*	*S. frugiperda*	*H. armigera*	*M. seperata*
1 × 10^8^ conidia/mL	CBM1	1.29 (0.88–1.63)	2.21 (1.69–2.69)	1.31 (0.88–1.67)	1.92 (0.25–3.12)
	CBM2	2.57 (1.87–3.22)	3.98 (3.26–4.78)	3.78 (3.13–4.43)	-
	CBM3	1.49 (1.05–1.87)	2.59 (2.01–3.13)	1.70 (1.11–2.20)	-
1 × 10^7^ conidia/mL	CBM1	2.09 (1.46–2.65)	2.48 (1.92–3.01)	1.68 (1.23–2.08)	-
	CBM2	3.41 (2.57–4.38)	4.91 (4.11–6.13)	4.87 (4.18–5.82)	-
	CBM3	3.22 (2.52–3.92)	2.97 (2.28–3.68)	2.94 (2.18–3.72)	-

**Table 2 insects-16-00287-t002:** LC_50_ of three *B. bassiana* isolates to four Lepidoptera larvae. LC_50_ denotes the lethal concentration to cause 50% mortality, determined on 7th day post infection with various spore concentrations using the soaking method. Each group contained 12 larvae and was performed in triplicate.

Strains	LC_50_ (1 × 10^5^ Conidia/mL) (95% Confidence Interval)			
	*P. xylostella*	*S. frugiperda*	*H. armigera*	*M. seperata*
CBM1	14.76 (0.03–58.67)	10.06 (0.11–39.71)	8.53 (0.21–36.34)	126.84
CBM2	47.82 (0.01–273.88)	52.10 (0.33–251.07)	39.17 (0.66–196.46)	-
CBM3	17.13 (0.03–67.87)	16.17 (0.19–66.59)	13.52 (0.313–61.99)	-

The values were calculated using the PROBIT analysis provided by IBM SPSS 26.0 at 95% confidence intervals.

## Data Availability

The data sets analyzed during this study are available from the corresponding author upon reasonable request.

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
