# Peer review of "Evaluating *Beauveria bassiana* Strains for Insect Pest Control and Endophytic Colonization in Wheat"

_insects, 2025, doi:10.3390/insects16030287_

Round 1
Reviewer 1 Report
Comments and Suggestions for Authors
The manuscript submitted by Liu et al. presents experimental data to show the virulence of three B. bassiana strains against four lepidopteran pests and the endophytic activity of a selected strain in wheat seedlings. Overall, the experiments were properly carried out, generating robust data for analysis. It is interesting to see that the selected strain enabled to colonize the wheat seedlings through soil drench after seeding and that the colonized seedlings grew better and showed insecticidal activity after ingestion by insects, suggesting the endophytic activity of B. bassiana to favor wheat growth and pest control. I recommend the manuscript to be considered for acceptance after a major revision.
Major concerns:
1. Delete Figure 1A-F, which is not informative for identification of the fungal isolates. Reorganize the rest panels. Do not italicize ‘sp.’
2. The concentration of conidial suspension above 108 conidia/mL is too high to be used in insect bioassays and also not applicable in practice. For this reason, panels in Figure 2 and Figure S1 can be combined to show survival trends only at 107 and 108 conidia/mL of each strain. Consequently, Table 1 and Table S1 can also be combined to present the LT50 values (with 95% CI) of each strain to each insect at the two spore concentrations, respectively.
3. The bioassay data used for the estimation of LC50 is questionable. For an ideal bioassay, gradient spore concentrations are tested to cause a range of insect mortality from ~5% to ~95%. The resultant inverted sigmoid mortality trend over the spore concentrations requires PROBIT analysis for the estimation of LC50 with 95% CI. In the study, however, the used spore concentrations ranged from 106 to 109 conidia/mL, resulting in a mortality trend not qualified (deficient of data points of low to median mortality) for conventional probit analysis. Therefore, I suggest the gradient concentrations of 104, 105, 106, 107 and 108 conidia/mL to be tested again in the bioassays. Otherwise, the part of LC50 estimation should be deleted from the manuscript.
4. Figure 3A does not like an image of control treatment. The control image should show the result of leaf pieces incubated after the same surface treatment but not colonized by the fungus before the surface treatment. In Figure 3B. the fungal colony grown from the colonized leaf piece should be enlarged for details, followed by a microscopic image for a view of hyphae and conidia.
Minor:
L19, delete ‘isolated’ and ‘that’.
L28, Change to ‘three B. bassiana strains.
L54, spell out ‘H. armigera’ here.
L55, change ‘yields’ to production.
L155-157, delete these lines including the formula. It is conventional to determine the concentration of conidial suspension with a hemocytometer.
Reviewer 2 Report
Comments and Suggestions for Authors
Title: Since the expression pest control is very broad (a pest includes animals, plants, microorganisms, among others), I suggest to delimit the title of the article.
Line 18: In this case, ¿What relevance does it have to include the species P. xylostella in this study?
Line 71: Please, update the fungal taxonomy.
Lines 96-100: Please add references.
Line 119: ¿Were the fungal strains deposited in a Collection? For example: ¿ARSEF or others? Please, provide the access numbers.
Lines 126-146: Please, combine the subsections 2.2.1, 2.2.2 and 2.2.3 into one: DNA extraction, PCR amplification and sequencing. It is not correct to use the expression “cloning” to refer to PCR amplification. They are different techniques.
Line 134: Please provide details about the program used in PCR experiments.
Lines 132-137: the authors do not provide information about the procedure used for the purification of the PCR products obtained. Please, provide it.
Line 138: ¿Were the sequences submitted to the GenBank database? Please, indicate accession numbers.
Line 162: ¿How many replicates were performed for each experiment? The researchers give information just about repetitions.
Lines 175-181: I think it is necessary to performed preliminary tests for each lepidopteran species, since the susceptibility could not be the same for each lepidopteran species. Please provide details about number of replicates performed.
Lines 183-191: Please, combine the subsections into one.
Lines 261-265: Please delete this paragraph. This information has already been included in materials and methods, as appropriate.
Lines 270-272: The morphological features are not enough to determine the fungal specie.
Lines 273-275: Please delete the sentence. This information has already been included in materials and methods, as appropriate.
Lines 276-281: Unfortunately, the data are not sufficient to confirm the identity of the fungal strains. Microscopic characterization is lacking, the photos are not of good quality and only conidia are observed. In addition, within the Beauveria genus, ITS sequences are not adequate to determine the species. For this, it is necessary to use other molecular markers. Please see Imoulan et al. 2016. Lastly, the bioinformatic analyses performed do not include the reference sequences of the all Beauveria species.
Lines 293-295: It has been demonstrated worldwide that entomopathogenic fungi act on contact. Please, provide references about the following affirmation: “or being ingested to exert their effects within the gut”.
Round 2
Reviewer 1 Report
Comments and Suggestions for Authors
This manuscript has been revised in proper response to my comments. I am satisfied with the authors' responses and revision and hence recommend the updated version for acceptance.